# Political violence and mental health outcomes among Palestinians: The mediating roles of dehumanization and humiliation

Dana Bdier[1] ⓘ, Fayez Mahamid[1] ⓘ, Bilal Hamamra[1] ⓘ and Guido Veronese[2] ⓘ

[1]An-Najah National University, Palestinian Territory, Occupied and [2]Università degli Studi di Milano-Bicocca, Italy

## Research Article

political violence; dehumanization; humiliation; mental health; Palestine

**Corresponding author:**
Dana Bdier;
Email: d.bdair@najah.edu

## Abstract

This study examines the mediating roles of dehumanization and humiliation in the relationship between political violence and mental health outcomes characterized by depression, anxiety and stress among Palestinians. This cross-sectional quantitative study was conducted in October 2024 with 633 Palestinian adults from the West Bank. The sample was recruited online through convenience sampling. Participants completed Arabic versions of the *Exposure to Political Violence Scale*, the *Experience of Dehumanization Scale*, the *Humiliation Inventory* and the *Depression, Anxiety and Stress Scale-21*. All measures were culturally adapted and validated. Ethical approval was obtained from the An-Najah National University, and informed consent was obtained. The findings revealed that political violence is positively associated with stress ($r = 0.38$), anxiety ($r = 0.35$) and depression ($r = 0.34$; all $p < 0.01$). Additionally, structural equation modeling revealed that political violence predicted higher stress ($\beta = 0.66$), anxiety ($\beta = 0.83$) and depression ($\beta = 0.77$), with significant indirect effects through dehumanization and humiliation ($\beta$ range = 0.21–0.28; $p < 0.01$). Findings highlight the strong associations between exposure to political violence and poorer mental health, particularly when accompanied by experiences of humiliation and dehumanization. This research highlights the importance of developing culturally tailored, community-based mental health programs in Palestine that address the psychological effects of these experiences and promote resilience and recovery.

## Impact statements

This study advances understanding of how political violence affects Palestinians' mental health by identifying dehumanization and humiliation as key mechanisms linking exposure to violence with depression, anxiety and stress. By showing that these experiences significantly mediate the impact of political violence, the findings highlight how harm extends beyond exposure to direct and physical traumatic events to include threats to dignity, identity and social value. The results have important implications for mental health practice and policy in Palestine. Interventions should focus on enhancing psychological resilience, *sumud* as a resilient resistance and promoting positive coping strategies for managing stress related to political violence. Moreover, there is a necessity to develop comprehensive community-level therapeutic interventions. For researchers, the study underscores the need to further examine the association between the study variables in areas affected by war.

## Theoretical background

The ongoing Israeli-Palestinian conflict has lasted for 74 years since Al-Nakbah, "the catastrophe," happened in 1948. Since then, Palestinians were exposed to the denial of their culture, as over 700,000 Palestinian Arabs – nearly half of prewar Palestine's Arab population – were forced to flee or were expelled from their homes and became refugees living in camps in neighboring countries such as Lebanon, Jordan and Syria, as well as in the West Bank and Gaza Strip (Manna', 2013; Caplan, 2019; Mahamid, 2020).

Palestinians have been exposed to different traumatic circumstances as a result of the long-standing Israeli military occupation, such as witnessing mutilated bodies on TV, hearing shelling of the area by artillery, seeing evidence of shelling and hearing sonic bombs from jetfighters or drones flying over the area (Thabet et al., 2014).

Political violence has devastating effects on mental health, particularly in regions like Palestine (Saeedi et al., 2025), where the population is subjected to chronic conflict, military occupation and systemic oppression. The psychological toll of political violence is far-reaching, impacting both individual and collective mental health. Shukri et al. (2022) found that Palestinians are much more likely to suffer from post-traumatic stress disorder (PTSD) and depression than the global average and their neighboring countries. The global prevalence of depression and PTSD are 5% and 3.6%, respectively. The prevalence of severe PTSD in children living in the Gaza Strip is

32.7%. Furthermore, depression in Palestine is among the highest rates in the world, affecting 40% of Palestinians. Moreover, Mahamid et al. (2021) found that traumatic life events were negatively associated with psychological well-being among Palestinians living in the West Bank. Furthermore, a study reviewed 37 studies examining the well-being of Palestinian individuals, focusing on the effects of conflict and war on their physical, mental and social health. PTSD, depression and anxiety were found to be among the most common mental health outcomes as a result of being exposed to political violence (Topuzb and Aslan, 2025). In addition, economic pressure related to the political instability was found to be associated with self-harm among Palestinian adults (Hamamra et al., 2025).

Central to understanding these outcomes are the mediating roles of dignity and humiliation, which play a critical part in how violence is experienced and how it shapes long-term mental health outcomes (Elbedour et al., 2007; Giacaman et al., 2011; Hobfoll et al., 2011; Barber et al., 2016). Political violence in Gaza is not confined to physical acts of aggression but involves a continuous effort to dehumanize Palestinians, stripping them of their dignity and inflicting profound psychological harm (Giacaman et al., 2007; Khamis, 2012).

## Dehumanization and the erosion of dignity

Dehumanization is a central feature of political violence, where individuals are systematically reduced to sub-human status to justify acts of violence against them (Hicks, 2011; Hartling and Lindner, 2018). In Palestine, this process is evident both in rhetoric and action. Israeli officials have frequently used dehumanizing language to frame Palestinians as dangerous, violent and undeserving of compassion or protection (Jabareen, 2010). Israeli Defense Minister Yoav Gallant's statement referring to Palestinians as "human animals" (Gallant, 2023) exemplifies this dehumanizing rhetoric. Such language not only legitimizes violence but also inflicts psychological harm (Bar-Tal, 2007; Elbedour et al., 2007).

This aligns with Giorgio Agamben's (1998) concept of *Homo Sacer*, in which certain groups are excluded from legal and moral protections, rendering their lives expendable (Cohen, 2016). Palestinians, like Agamben's *Homo Sacer* (1998), are placed in a state of exception, where violence against them is normalized, and their lives are treated as disposable (Hajjar, 2005). This systematic dehumanization reinforces feelings of powerlessness and insecurity among Palestinians living under prolonged occupation (Hammami, 2004; Giacaman et al., 2011).

Dehumanization also has profound effects on the collective consciousness of Palestinians. When an entire community is treated less than humans, individuals feel internally that they are less than others and worthless (Rouhana and Bar-Tal, 1998). This collective experience of dehumanization creates a sense of shared trauma, reinforcing feelings of worthlessness and helplessness (Hammami, 2004). Examples of such practices against Palestinians in the West Bank are the demolition of houses; the acquisition of Palestinian land by force; separating neighborhoods and villages by checkpoints, and the establishment of the Separation Wall; collective punishment such as detention and torture; and forcible transfer of Palestinians (Ladadwa and Nasr, 2022).

## Humiliation as a tool of political violence

Humiliation is a critical psychological weapon in political violence, particularly in conflict zones like Gaza. The experience of humiliation undermines an individual's sense of dignity, leading to long-term psychological damage (Lindner, 2006). Palestinians are subjected to humiliation not only through direct violence but also through the daily degradation imposed by the occupation (Giacaman et al., 2007). Acts of humiliation include the destruction of homes, the denial of medical care and the restriction of movement, all of which are designed to diminish the dignity of the population (Barber, 2000; McNeely et al., 2015).

Humiliation is not only experienced individually but collectively, as entire communities face systemic degradation (B'Tselem, 2021). This collective humiliation is strongly linked to the development of mental health disorders, including PTSD, depression and anxiety (Giacaman et al., 2011; Khamis, 2012). Humiliation weakens the protective factors, reinforces psychological resilience and leaves them more vulnerable to trauma (Hartling and Lindner, 2018).

Exposure to humiliation during childhood has been found to have long-term psychological effects on Palestinians, contributing to the development of behavioral problems, anxiety and trauma-related disorders (Punamaki et al., 2005; Qouta et al., 2008). The experience of watching one's family members be humiliated or degraded by occupying forces creates a deep psychological scar that can persist into adulthood (Elbedour et al., 2007).

## Restoring dignity and the role of collective trauma

Dignity is essential for psychological well-being, and its violation has significant mental health consequences (Hicks, 2015; Kteily et al., 2015). In Palestine, the continuous violation of dignity through dehumanization and humiliation plays a central role in shaping mental health outcomes. The destruction of homes, restrictions on movement and the denial of basic human rights erode the dignity of the population, contributing to feelings of helplessness, worthlessness and powerlessness (Giacaman et al., 2011; Hobfoll et al., 2011).

Research has shown that the restoration of dignity is critical for psychological recovery (Hicks, 2015; Hartling and Lindner, 2018). Social support, personal resistance and community solidarity can help in fostering agency (Hobfoll et al., 2012).

The collective loss of dignity also plays a significant role in shaping the mental health outcomes of the Palestinians through generations. The shared experience of humiliation and degradation creates a communal sense of worthlessness and despair, which contributes to the development of mental health disorders on a larger scale (Giacaman et al., 2007; Barber et al., 2016). This collective trauma is particularly damaging in a cultural context where respect, honor and social standing are central to identity and psychological well-being (Barber, 2000; Hajjar, 2005).

## Global indifference and perpetuation of trauma

The international community's relative inaction regarding the violence in Palestine exacerbates the psychological damage caused by political violence. The global failure to hold Israel accountable for its actions sends a message to Palestinians that their lives and rights are of lesser value, further reinforcing the dehumanization they experience (Aswadi, 2023). This global indifference not only allows the violence to continue but also deepens the feelings of humiliation and abandonment, compounding the psychological harm caused by the conflict (Giacaman et al., 2011; Hobfoll et al., 2011).

Media coverage also plays a role in reinforcing global indifference. Western media often frames the Israeli-Palestinian conflict in ways that dehumanize Palestinians, portraying Israeli military actions as self-defense while downplaying the humanitarian crisis in Palestine (Fabian, 2023). This biased portrayal further marginalizes Palestinian suffering and reinforces the perception that their lives are less valuable (Shalal and Singh, 2023).

Addressing the psychological effects of global indifference requires a shift in how the international community engages with the conflict in Palestine. Greater recognition of Palestinian suffering, coupled with a commitment to upholding international law, is essential for restoring dignity to those affected by the violence (Hobfoll et al., 2012). Without meaningful global intervention, the cycle of humiliation and psychological trauma will continue, deepening the mental health crisis in Palestine (Hajjar, 2005).

### Current study

This study investigates whether political violence predicts stress, depression and anxiety among Palestinians, with a focused analysis on the mediating roles of humiliation and dehumanization between these variables. This research is groundbreaking, as to our knowledge, it is the first to explore the relationship between these variables within the Palestinian context. Based on prior studies (Hammami, 2004; Punamaki et al., 2005; Qouta et al., 2008; Giacaman et al., 2011; Hobfoll et al., 2011; Barber et al., 2016; Hartling and Lindner, 2018), the study hypotheses were defined as follows: First (H1), political violence would be positively associated with depression, anxiety and stress; and second (H2), humiliation and dehumanization would mediate the associations between political violence and these mental health outcomes.

### Methods

#### Participants and procedures

The research was conducted in October 2024 and targeted Palestinians living in the West Bank of Palestine. Participants were recruited using online methods, including emails, social media and advertisements. The aims of the study were presented online, and participants interested in participating were asked to send an email indicating their willingness to join the study. All participants received a letter clarifying the objectives and ethical issues of the study. They provided written informed consent upon accepting the conditions of participation. A total of 633 adults participated in the study, comprising 310 males and 323 females. Of the participants, 58.8% were from urban regions, 36.7% were from rural regions and the remaining 4.5% were from internally displaced camps. Regarding educational attainment, 26.4% held a graduate degree, 65.9% held a bachelor's degree and 7.7% held a high school degree. To be included in the study, participants were required to be (1) native Arabic speakers, (2) Palestinian and (3) residents in the occupied Palestinian territories (oPt). Approval for the study was obtained from An-Najah National University Institutional Review Board before data collection began.

#### Measures

Following standard methodological recommendations for questionnaire development (Hambleton et al., 2005), all measures not already validated in Arabic were translated and back-translated from the original English version into Arabic. This process involved a panel of 10 Arab professionals in psychology, counseling and social work, who evaluated the clarity and relevance of the questions and translations. After completing the initial draft of translated items, the questionnaires were back-translated into English by an independent expert English editor. Based on their feedback, the translated version was pilot-tested among 80 participants and further refined for clarity.

#### The experience of dehumanization scale

This scale aims to assess how individuals perceive their own dehumanization in various contexts. It includes items that capture different dimensions of dehumanization, such as feelings of being treated as less than human, experiencing a lack of empathy from others and the impact of these experiences on one's self-identity and well-being. Participants rated how often they experienced dehumanization, as represented by each of the 10 items in the past 6 months, on a 5-point Likert-type scale ranging from 1 (*never*) to 5 (*all the time*).

#### Exposure to *Political* Violence Scale *(EPVS)*

The EPVS (Haj-Yahia, 2005) examines 49 acts of political violence, which reflect experience with psychological abuse, physical violence, sexual abuse (as a direct victim) and witnessing of such violence (as an indirect victim) committed by the Israeli army and police, as well as by Israeli settlers against family members and relatives. Participants were requested to answer the question about each act using a dichotomous measure: no = 0 or yes = 1. Cronbach's alpha value of the EPVS, as used in the females' study, was 0.90, while in the males' study it was 0.91.

#### Depression, Anxiety *and* Stress Scale *(DASS-21)*

The DASS-21 is a 21-item self-report questionnaire designed to measure the severity of a range of symptoms common to both depression and anxiety. In completing the DASS, the individual is required to indicate the presence of a symptom over the previous week. Each item is scored from 0 (*did not apply to me at all over the last week*) to 3 (*applied to me very much or most of the time over the past week*). The essential function of the DASS is to assess the severity of the core symptoms of depression, anxiety and stress. Accordingly, the DASS allows not only a way to measure the severity of a patient's symptoms but also a means by which a patient's response to treatment can be measured. The scale to which each item belongs is indicated by the letters D (depression), A (anxiety) and S (stress). For each scale (D, A and S), sum the scores for identified items. Because the DASS 21 is a short form version of the DASS (the long form has 42 items), the final score of each item group (depression, anxiety and stress) needs to be multiplied by 2 (×2) (Gomez, 2016).

#### Humiliation *inventory (HI)*

The HI measures humiliation along two subscales: fear of humiliation and cumulative (Hartling and Luchetta, 1999). This 32-item scale is divided into four sections, the first of which assesses how much participants believe themselves to have been affected by particular experiences ("Throughout your life, how seriously have you felt harmed by being ridiculed?"). The second section measures how fearful participants are of being humiliated ("At this point in your life, how much do you fear being harassed?"). The next section attempts to measure participants' concerns over experiencing humiliation ("At this point in life, how concerned are you about being discounted as a person?"). The last section consists of only two items, which assess participants' worries ("How worried

are you about being viewed by others as incompetent?"). All 32 items are measured on a 5-point Likert scale, ranging from 1 ("*Not at all*") to 5 ("*Extremely*") (Asmari et al., 2022).

### Data analysis

We used descriptive statistics to examine the main characteristics of the study variables and tested the correlations between these variables: political violence, humiliation, dehumanization, stress, anxiety and depression. Structural equation modeling (SEM) was employed to assess the conceptual model (see Figure 1) of our study, where political violence operated as the predictive variable. Humiliation and dehumanization served as mediating variables, while mental health outcomes (stress, depression and anxiety) were the outcome variables. The model demonstrated good fit indices: Comparative Fit Index (CFI) = 0.99, Normed Fit Index (NFI) = 0.98, Incremental Fit Index (IFI) = 0.98, Standardized Root Mean Square Residual (SRMR) = 0.03 and Root Mean Square Error of Approximation (RMSEA) = 0.04. The SEM model (see Figure 2) was tested using AMOS 29 statistical analysis software.

### Findings

Descriptive statistics for political violence, stress, anxiety, depression, humiliation and dehumanization are presented in Table 1. Mean scores and standard deviations (SDs) were as follows: political violence ($M = 1.51$, SD = 0.25), stress ($M = 3.23$, SD = 0.76), anxiety ($M = 2.83$, SD = 0.76), depression ($M = 3.21$, SD = 0.74), humiliation ($M = 2.16$, SD = 0.88) and dehumanization ($M = 4.71$, SD = 0.46). All measures demonstrated a high degree of reliability, with coefficients ranging from 0.87 for humiliation to 0.94 for stress. To better interpret the findings of our study, we adopted a scoring key based on the 1–5 Likert scale: 1.00–2.00 = *low*, 2.01–3.00 = *moderate*, 3.01–4.00 = *high* and 4.01–5.00 = *very high* (see

Table 1). According to this key, political violence was low, stress and depression were high, anxiety and humiliation were moderate and dehumanization was very high. In addition, all measures used in this study demonstrated a high degree of reliability, with coefficients ranging from 0.87 for Humiliation to 0.94 for stress.

Results of the correlational analysis, presented in Table 2, revealed several significant relationships. Political violence was positively correlated with stress ($r = 0.38$, $p < 0.01$), anxiety ($r = 0.35$, $p < 0.01$), depression ($r = 0.34$, $p < 0.01$), humiliation ($r = 0.11$, $p < 0.05$) and dehumanization ($r = 0.22$, $p < 0.01$). Stress showed positive correlations with anxiety ($r = 0.86$, $p < 0.01$), depression ($r = 0.82$, $p < 0.01$), humiliation ($r = 0.34$, $p < 0.01$) and dehumanization ($r = 0.24$, $p < 0.01$). In addition, anxiety was positively correlated with depression ($r = 0.76$, $p < 0.01$), humiliation ($r = 0.31$, $p < 0.01$) and dehumanization ($r = 0.21$, $p < 0.01$). Moreover, depression was positively correlated with humiliation ($r = 0.46$, $p < 0.01$) and dehumanization ($r = 0.30$, $p < 0.01$). Finally, humiliation was positively correlated with dehumanization ($r = 0.16$, $p < 0.05$).

### Structural equation modeling

The results of the SEM analysis are illustrated in Figure 2. The hypothesized model, depicted in Figure 1, includes political violence as a predictor, with humiliation and dehumanization as mediating variables. The outcomes assessed in this model are mental health outcomes (stress, depression and anxiety).

Analysis of the paths showed positive effects between political violence and humiliation ($\beta_{M, Y} = 0.39$; $p < 0.01$), dehumanization ($\beta_{M, Y} = 0.41$; $p < 0.01$), stress ($\beta_{M, Y} = 0.66$; $p < 0.01$), depression ($\beta_{M, Y} = 0.77$; $p < 0.01$) and anxiety ($\beta_{M, Y} = 0.83$; $p < 0.01$). In addition, results of path analysis showed positive effects between humiliation and stress ($\beta_{M, Y} = 0.25$; $p < 0.01$), anxiety ($\beta_{M, Y} = 0.26$; $p < 0.01$) and depression ($\beta_{M, Y} = 0.34$; $p < 0.01$). Moreover, results of path analysis showed positive effects between dehumanization and stress

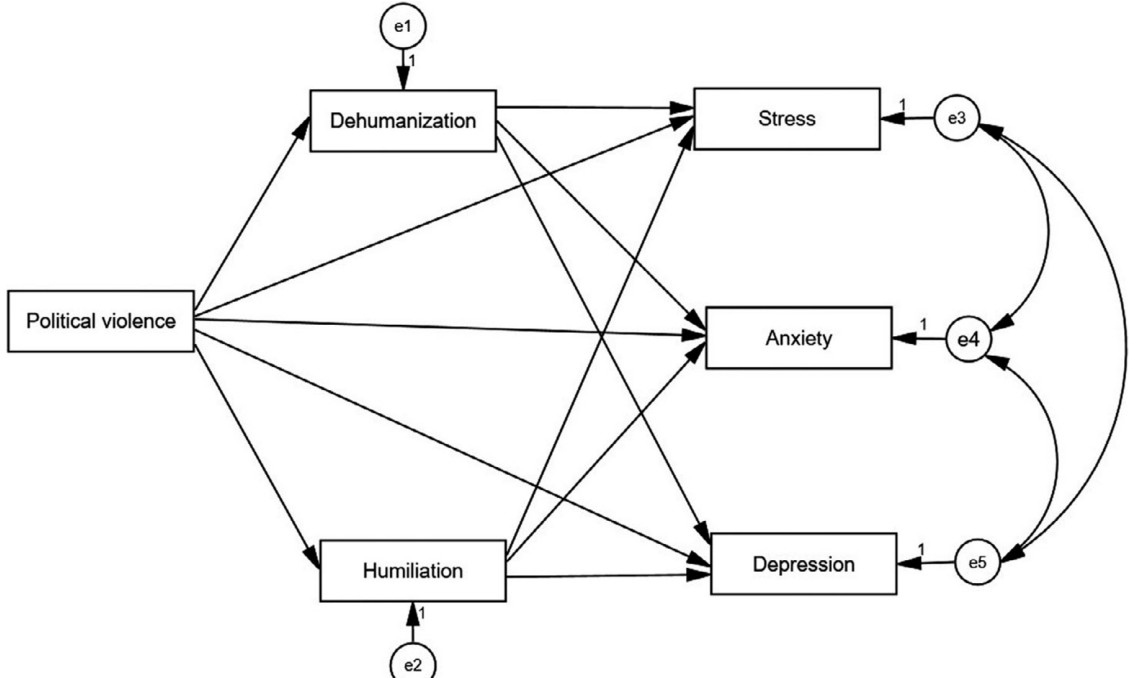

**Figure 1.** The conceptual effect of political violence on mental health outcomes, and the mediating roles of humiliation and dehumanization.

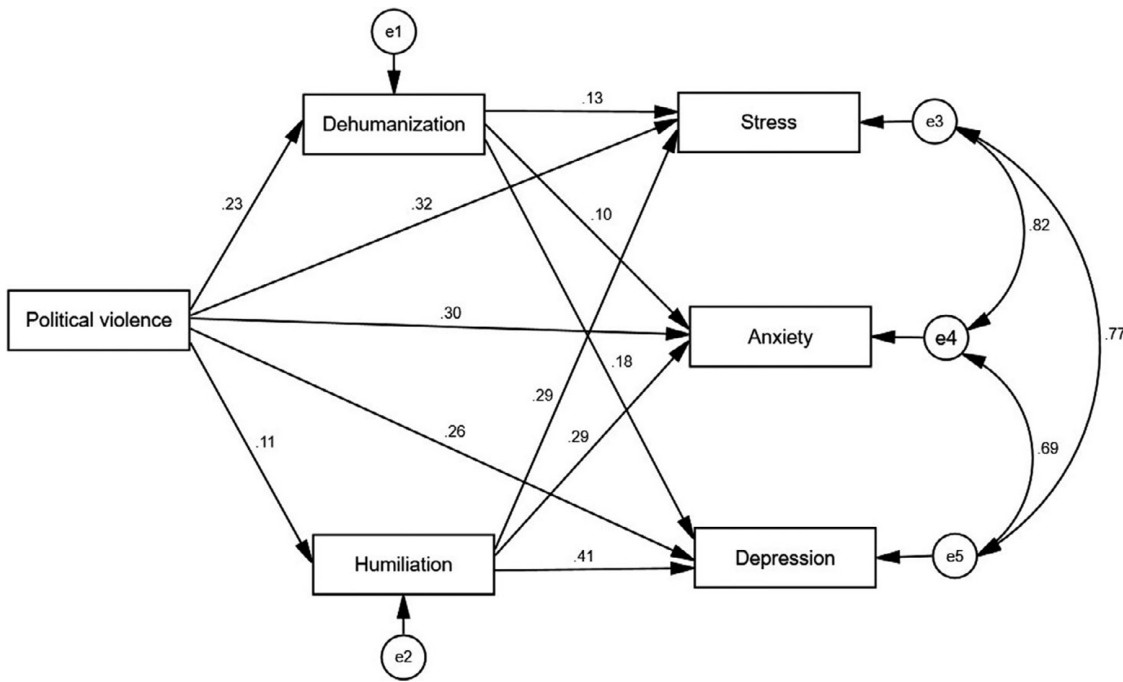

**Figure 2.** Structural equation modeling of political violence on mental health outcomes, and the mediating roles of humiliation and dehumanization.

**Table 1.** Descriptive statistics for research variables ($N = 633$)

| Variable | M | SD | Min | Max | Range | Skewness | Kurtosis | Cronbach's alpha |
|---|---|---|---|---|---|---|---|---|
| Political violence | 1.51 | 0.25 | 1.00 | 2.00 | 1.00 | −0.07 | 0.69 | 0.92 |
| Stress | 3.23 | 0.76 | 2.00 | 5.00 | 3.00 | 0.41 | 0.62 | 0.94 |
| Anxiety | 2.83 | 0.76 | 2.00 | 5.00 | 3.00 | 0.50 | 0.53 | 0.92 |
| Depression | 3.21 | 0.74 | 2.00 | 5.00 | 3.00 | 0.27 | 0.63 | 0.89 |
| Humiliation | 2.16 | 0.88 | 1.00 | 4.83 | 3.83 | 0.83 | 0.28 | 0.87 |
| Dehumanization | 4.71 | 0.46 | 2.60 | 5.00 | 2.40 | 0.305 | 0.85 | 0.90 |

**Table 2.** Correlations among study variables ($N = 633$)

| Measures | 1 | 2 | 3 | 4 | 5 | 6 |
|---|---|---|---|---|---|---|
| Political violence | 1 | 0.38** | 0.35** | 0.34** | 0.11** | 0.22** |
| Stresses | | 1 | 0.86** | 0.82** | 0.34** | 0.24** |
| Anxiety | | | 1 | 0.79** | 0.33** | 0.21** |
| Depression | | | | 1 | 0.46* | 0.30** |
| Humiliation | | | | | 1 | 0.16* |
| Dehumanization | | | | | | 1 |

**$\alpha$ is significant at ≤ 0.01.
*$\alpha$ is significant at ≤ 0.05.

($\beta_{M, Y} = 0.21$; $p < 0.01$), anxiety ($\beta_{M, Y} = 0.15$; $p < 0.05$) and depression ($\beta_{M, Y} = 0.28$; $p < 0.01$).

Concerning the mediating hypothesis, our model revealed a standardized total effect of political violence on depression ($\beta_{X, M} = 0.77$; $p < 0.001$), with significant indirect effect via humiliation ($\beta_{X, M, Y} = 0.26$; $p < 0.01$) and dehumanization ($\beta_{X, M, Y} = 0.27$; $p < 0.01$). The model also revealed a standardized total effect of political violence on anxiety ($\beta_{X, M} = 0.83$; $p < 0.001$), with

significant indirect effect via humiliation ($\beta_{X, M, Y} = 0.28$; $p < 0.01$) and dehumanization ($\beta_{X, M, Y} = 0.22$; $p < 0.01$). Finally, the model revealed a standardized total effect of political violence on stress ($\beta_{X, M} = 0.66$; $p < 0.001$), with significant indirect effect via humiliation ($\beta_{X, M, Y} = 0.21$; $p < 0.01$) and dehumanization ($\beta_{X, M, Y} = 0.24$; $p < 0.01$).

## Discussion

This study investigates the relationship between political violence and mental health outcomes – namely stress, anxiety and depression – among Palestinians, with a focus on how humiliation and dehumanization mediate these effects. The results revealed a significant association between exposure to political violence and increased psychological distress, where humiliation and dehumanization acted as crucial mediators. The findings highlight the complex mental health impact of sustained political violence, intensified by the systematic dehumanization and humiliation Palestinians endure under the Israeli occupation.

The association between political violence and poor mental health is well-documented in conflict studies. Research by Johnson and Thompson (2008) found that ongoing violence and instability

strongly predict trauma-related symptoms, a trend similarly observed by Hall et al. (2015) and Steel et al. (2009), who identified exposure to violence as a major contributor to mental health disorders. For Palestinians, these findings resonate with their daily reality: constant threats to life and well-being, limitations on movement and recurrent confrontations with the Israeli military forces. These conditions create a sustained sense of insecurity, contributing to elevated levels of stress, depression and anxiety (Giacaman et al., 2011; Hobfoll et al., 2011; Barber et al., 2016).

The mediating roles of humiliation and dehumanization in this study provide insight into how political violence under Israeli military occupation impacts Palestinians' mental health. Dehumanization, for instance, is not only a consequence of political violence but also a calculated mechanism within it. Palestinians are frequently subjected to language and treatment that reduce their status to less than human, which erodes their inherent worth (Hicks, 2011; Hartling and Lindner, 2018). This dehumanizing rhetoric has been widely observed in official statements and media, with some figures labeling Palestinians as "human animals" or "threats," a characterization that legitimizes and even normalizes violence against them. This type of rhetoric not only serves to justify political violence but also reinforces a profound sense of worthlessness and disposability among those affected (Bar-Tal, 2007; Elbedour et al., 2007).

The theoretical framework of *Homo Sacer*, proposed by Agamben (1998), helps contextualize these experiences, where Palestinians are placed in a "state of exception," essentially stripped of protections that are afforded to others (Mayaleh et al., 2024). This status, legally and socially, treats Palestinians as expendable and deprives them of the security that is fundamental to psychological well-being (Hajjar, 2005; Cohen, 2016). Without legal recognition of their suffering, Palestinians face unique challenges in coping with the psychological toll of such violence. The constant reinforcement of their status as "other" impacts their self-perception, increasing levels of depression, anxiety and hopelessness (Hammami, 2004; Giacaman et al., 2007).

Humiliation, similarly, aggravates the mental health impacts of political violence. Lindner (2006) and McNeely et al. (2015) demonstrate that prolonged exposure to humiliation in everyday life can undermine resilience, leading to long-lasting psychological damage. In Gaza and the West Bank, humiliation is often deliberately inflicted through Israeli actions like home demolitions, invasive checkpoints and the denial of medical resources. The psychological harm of these experiences shapes a collective identity that internalizes shared suffering, leading to what Rouhana and Bar-Tal (1998) described as "collective trauma." This collective trauma is especially harmful to future generations, as evidenced by studies of Palestinian children exposed to political violence, who often display signs of anxiety, depression and emotional distress (Punamaki et al., 2005; Qouta et al., 2008). By compounding each individual's sense of helplessness, these experiences contribute to a wider atmosphere of community despair and helplessness.

Beyond the immediate psychological consequences, the study's findings also suggest that humiliation and dehumanization limit recovery options. In the Palestinian territories, the blockade, lack of mental health services and restrictive movement policies hinder these processes. Without access to therapeutic resources and protective measures, individuals and communities face cycles of trauma, which reinforce anxiety, stress and depressive symptoms. Keys et al. (2014) further point out that the compounded effects of perceived dehumanization and humiliation exacerbate barriers to healthcare access, perpetuating trauma and distress.

These findings highlight the urgent need for international advocacy and mental health interventions that focus on cultural resilience and community support within the Palestinian context. This may include community-led psychosocial support, training local community health providers and integrating positive religious coping within existing social networks to enhance individuals' and communities' capacity to deal with ongoing political conflicts.

This recommendation is supported by Punamaki et al. (2005) study, which highlights the importance of culturally sensitive mental health services and community solidarity in regions experiencing prolonged conflict. Without such initiatives, Palestinians will likely continue to experience high levels of psychological distress, exacerbated by structural violence and dehumanizing rhetoric that strips them of both individual and collective dignity.

## Limitations

Our study has several limitations that could inform future research. First, it focused primarily on Palestinians residing in the West Bank. Future studies should investigate the effects of political violence on mental health in other areas of Palestine, such as East Jerusalem and the Gaza Strip, to gain a broader understanding of how political violence impacts mental health across different Palestinian contexts. Second, our research employed a cross-sectional quantitative design to examine the relationships between political violence, stress, depression and anxiety, as well as the mediating effects of humiliation and dehumanization. Future research should consider adopting mixed methods to explore these variables and additional relevant factors in greater depth, providing a more comprehensive understanding of the findings. Finally, it would be beneficial to test the psychometric properties of humiliation and dehumanization scales, as these tools have not yet been validated within the Palestinian context.

## Conclusion

The current study aimed to examine the relationship between exposure to political violence and mental health outcomes, specifically depression, anxiety and stress. Our results showed a positive correlation between exposure to political violence and increased levels of stress, depression and anxiety. The findings further indicated that humiliation and dehumanization served as mediators in the relationship between exposure to political violence and these mental health outcomes. The novelty of the study lies in testing the role of humiliation and dehumanization as mediating variables in the relationship between political violence and mental health outcomes, as these variables have not previously been tested within the Palestinian context. These results draw attention to the continuous violence experienced by the Palestinian people, which correlates with rising levels of psychological stress, anxiety and depression within the community. Palestinians face significant political violence, marked by frequent incursions into cities, villages and refugee camps, resulting in ongoing collective trauma. The study also highlights the inhumane treatment and humiliation endured by Palestinians, which contributes to heightened anxiety and psychological distress.

The high rates of psychological stress, anxiety and depression among Palestinians necessitate the development of comprehensive community-level therapeutic interventions. These interventions should focus on enhancing psychological resilience, *sumud* as a resilient resistance, which means sticking to the land, as found to be

a crucial protective factor among Palestinians against exposure to political violence (Ryan, 2015), and promoting positive coping strategies for managing stress related to political violence. It is also crucial to target specific community groups within Palestinian society that may be especially vulnerable to political violence, such as residents of refugee camps and those living in areas frequently affected by Israeli military incursions. There is an urgent need for international mental health organizations operating in the Palestinian territories to strengthen the skills of Palestinian mental health professionals, enhancing their capacity to design and implement various mental health interventions. Moreover, and most importantly, than trying to support Palestinians and to boost their resilience by focusing on their individual and communal resources, it is crucial to recognize the structural injustice that Palestinians are going through, and to try to stop and remove it, as it is considered essential in boosting their resilience.

**Open peer review.** To view the open peer review materials for this article, please visit http://doi.org/10.1017/gmh.2026.10162.

**Data availability statement.** The datasets generated during and/or analyzed during the current study are available from the corresponding author on reasonable request.

**Author contribution.** All authors contributed equally to this work. All authors read and approved the final manuscript.

**Financial support.** No funding was received for this study.

**Competing interests.** The authors declare none.

**Ethics approval and consent to participate.** All procedures performed in this study involving human participants were in accordance with the ethical standards of An-Najah National University Institutional Review Board, the American Psychological Association (APA, 2010) and with the 2013 Helsinki Declaration. Informed consent was obtained from all participants.

**Informed consent.** Written informed consent was taken from all participants.

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
