## [Reviewer Report]

Thank you for such a great and relevant manuscript.

The study methodology, results, discussion, and conclusion are well done.

The limitations were appropriately acknowledged.

Please review and modify the keywords to better suit the study’s focus.

All references in the text and in the reference list should be checked for accuracy and proper formatting.

Incorporating recent Palestinian studies would further strengthen and enrich the manuscript.

---

## [Reviewer Report]

Report on Political violence

I have reviewed the article titled “Political Violence and Mental Health Outcomes among Palestinians: The Mediating Roles of Dehumanization and Humiliation” (Manuscript ID: GMH-2025-0296), submitted to Cambridge Prisms: Global Mental Health.

The manuscript investigates the relationship between exposure to political violence and mental health outcomes—specifically stress, anxiety, and depression—among Palestinians, with a focus on the mediating roles of humiliation and dehumanization. However, the manuscript would benefit from revisions, as detailed in the comments below.

Abstract

One limitation of the abstract is in the way the objectives are written. The wording is a bit repetitive, saying both “impact of political violence on mental health” and “influencing levels of depression, anxiety, and stress.” This repeats the same idea instead of adding clarity. It would be stronger to focus directly on how humiliation and dehumanization act as mediators, rather than restating the general impact of political violence.

The methods section is also vague. It mentions “validated questionnaires” but does not name the specific tools or explain if they were adapted for Palestinian culture. The abstract also leaves out how participants were chosen—whether by random sampling, convenience, or another method. In addition, no ethical procedures are mentioned, which is important in sensitive political contexts. Without these details, it is hard to judge how reliable or representative the findings are.

The results are presented in very general terms. Saying that political violence is “positively associated” with mental health problems is expected but does not provide much information. The use of structural equation modeling is noted, but the abstract does not explain what the results actually showed, such as key numbers or effect sizes. Adding even one or two statistics would make the results more convincing and informative.

The conclusions also raise concerns. The wording suggests causality, using phrases like “impact” and “consequences,” even though the study used a cross-sectional design, which cannot prove cause and effect. The recommendations are also too broad, mentioning only “targeted mental health interventions” without offering specific ideas, such as community-based or culturally tailored approaches. This makes the conclusions sound stronger than the data allow and less useful for practice.

Introduction

Authors are advised to add a concise Introduction before the theoretical background in order to frame the context (political violence in Palestine); show and highlight the research gap (little research on humiliation and dehumanization as mediators); and state the aim and contribution of the study early.

Theoretical background

The text repeatedly emphasizes the same points—especially about humiliation, dehumanization, and loss of dignity—without always adding new insights. This creates a sense of circular argumentation. For example, the idea that dehumanization strips dignity and causes psychological harm is stated in multiple places, sometimes with near-identical language. Authors need to cut repetitive statements and structure the background around two or three core concepts (e.g., humiliation, dehumanization, dignity), with collective trauma and intergenerational effects as extensions.

The section speaks almost exclusively about Gaza, while the study itself (from the abstract and methods) focuses on the West Bank. Since the data come from the West Bank, either broaden the theory to cover both West Bank and Gaza, or clarify why Gaza is emphasized.

The section focuses heavily on victimization and psychological harm but gives little space to resilience, coping strategies, community solidarity, or resistance—factors that many scholars (e.g., Barber, Hobfoll) also highlight. The lack of balance risks pathologizing Palestinians as passive victims and may be criticized as one-dimensional. Therefore, authors need to include literature on coping strategies, solidarity, and sumud to avoid portraying Palestinians only as traumatized victims.

Many references are older (2000s, early 2010s) or general (Silove, Lindner), and there is less engagement with the most recent trauma and dignity literature in global health and psychology. Authors need to incorporate recent literature (2022–2025) on trauma, dignity, and political violence in global health contexts. Authors are advised to refer to these significant recent works:

Reclaiming identity: The Gaza War’s role in shaping Palestinian university students' resilience and life’s meaning. Strong link to resilience, meaning-making, and psychological adjustment after political violence.

Understanding the Roots and Mechanisms of Self-Harm among Palestinian University Students: A Mixed-methods Study. Directly tied to mental health outcomes and coping mechanisms in youth.

Navigating health challenges: the interplay between occupation-imposed movement restrictions, healthcare access, and community resilience. Connects political violence/restrictions to health outcomes and resilience.

Political Socialization and Its Impact on Psychological Resilience and PTSD among Individuals Engaged with Israeli Occupation Forces on Mount Sabih. Explicitly deals with PTSD, resilience, and political conflict — very close to your theoretical framing.

Online education and its impact on university students’ mental health in the West Bank and Gaza Strip: a cross-sectional study. Focuses on mental health under occupation-related disruptions, relevant for contextualizing stress and anxiety.

Impact of Using Social Media on Facilitating Grassroots Mobilization and Activism Among Palestinians. Not about mental health directly, but useful for linking agency, coping, and collective resistance to dehumanization.

University students’ attitudes toward the stalled peace process and normalization with the Israeli occupation (2024-06-19)

The section covers dehumanization, humiliation, dignity, collective trauma, intergenerational trauma, global indifference, media bias—all important but sprawling. The lack of clear sub-organization risks overwhelming the reader. A tighter focus on key mediators (humiliation and dehumanization) with briefer mention of secondary themes would align better with the study’s purpose. Authors may divide the section into subsections (e.g., Dehumanization, Humiliation, Dignity and Mental Health, Collective Trauma) for clarity and flow.

Current study

The assertion that this is the first study of its kind may be overstated. Related research exists on mental health and political violence in Palestine, including studies on resilience, collective trauma, and psychosocial outcomes (e.g., Reclaiming identity: The Gaza War’s role in shaping Palestinian university students' resilience and life’s meaning, 2025; Navigating health challenges: the interplay between occupation-imposed movement restrictions, healthcare access, and community resilience, 2024). Acknowledging these studies would strengthen credibility.

While humiliation and dehumanization are central, the section does not justify why these specific mediators were chosen over other psychosocial factors such as resilience, social support, or collective trauma. Works such as Political Socialization and Its Impact on Psychological Resilience and PTSD (2024) or Online education and its impact on university students’ mental health in the West Bank and Gaza Strip (2024) illustrate other relevant mediators and could provide context.

The study could strengthen its contribution by situating the hypotheses within broader discussions of social, cultural, and political determinants of mental health in Palestine. For example, works like Navigating health challenges: the interplay between occupation-imposed movement restrictions, healthcare access, and community resilience (2024) or Reclaiming identity…resilience and life’s meaning (2025) highlight how structural and collective factors shape psychological outcomes and could inform interpretation.

Methods

Authors need to provide validation information for the dehumanization and humiliation scales in Palestinian populations.

Findings

Terms like “high,” “moderate,” and “average” are relative and not clearly defined with reference to the scale ranges. Providing exact means and standard deviations would give a clearer understanding of the distribution.

Discussion

Authors should use more cautious, correlational language instead of implying direct causation to reflect correlational findings (e.g., “humiliation and dehumanization were associated with increased psychological distress” instead of “act as crucial mediators”).

Some points—such as the impact of humiliation and dehumanization on mental health—are repeated multiple times, reducing conciseness.

Although prior studies on political violence and mental health are cited, little is said about whether prior research has or has not examined humiliation and dehumanization as mediators, which was the stated research gap.

While community-based programs are suggested, the discussion could outline concrete strategies or culturally adapted interventions.

Conclusion

Emphasize how this study fills the research gap on humiliation and dehumanization as mediators in Palestine.

Recommendations in the conclusion could include more concrete examples of “comprehensive community-level therapeutic interventions” or “positive coping strategies” to enhance applicability.

---

## [Reviewer Report]

Thank you for your revisions. I confirm that you have adequately addressed the comments I previously provided. The manuscript is now clearer, better structured, and the methodological details are more transparent. I appreciate your careful responses and the improvements made throughout the text.